# Predicting 30-day and 1-year mortality in heart failure with preserved ejection fraction (HFpEF)

**Ikgyu Shin**[1☯¤], **Nilay Bhatt**[1☯¤], **Alaa Alashi**[2], **Keervani Kandala**[1],
**Karthik Murugiah**[2,3*]

**1** Yale School of Public Health, New Haven, Connecticut, United States of America, **2** Section of Cardiovascular Medicine, Department of Internal Medicine, Yale School of Medicine, New Haven, Connecticut, United States of America, **3** Center for Outcomes Research and Evaluation, Yale-New Haven Hospital, New Haven, Connecticut, United States of America

☯ These authors contributed equally to this work.
¤ Current Address: Department of Biomedical Informatics and Data Science, Yale School of Medicine, New Haven, CT, United States of America
* karthik.murugiah@yale.edu

## Abstract

### Objectives

To develop and compare prediction models for 30-day and 1-year mortality in Heart failure with preserved ejection fraction (HFpEF) using EHR data, utilizing both traditional and machine learning (ML) techniques.

### Background

HFpEF represents 1 in 2 heart failure patients. Predictive models in HFpEF, specifically those derived from electronic health record (EHR) data, are less established.

### Methods

Using MIMIC-IV EHR data from 2008−2019, patients aged ≥ 18 years admitted with a primary diagnosis of HFpEF were identified using ICD-9 and 10 codes. Demographics, vital signs, prior diagnoses, and lab data were extracted. Data was partitioned into 80% training, 20% test sets. Prediction models from seven model classes (Support Vector Classifier (SVC), Logistic Regression, Lasso Regression, Elastic Net, Random Forest, Histogram-based Gradient Boosting Classifier (HGBC), and eXtreme Gradient Boosting (XGBoost)) were developed using various imputation and oversampling techniques with 5-fold cross-validation. Model performance was compared using several metrics, and individual feature importance assessed using SHapley Additive exPlanations (SHAP) analysis.

**Data availability statement:** The data that support the findings of this study are openly available in Physionet. Code used to analyze data and build the models is publicly accessible via [GitHub] (https://github.com/itsjustnilay/Predictive_Modelling_for_Heart_Failure_with_Preserved_Ejection_Fraction).

**Funding:** Dr. Murugiah received support from the National Heart, Lung, and Blood Institute of the National Institutes of Health (under award K08HL157727). The funders had no role in study design, data collection and analysis, decision to publish, or preparation of the manuscript.

**Competing interests:** The authors have declared that no competing interests exist.

## Results

Among 3,235 hospitalizations for HFpEF, 30-day mortality was 6.3%, and 1- year mortality was 29.2%. Logistic regression performed well for 30-day mortality (Area Under the Receiver operating characteristic curve (AUC) 0.83), whereas Random Forest (AUC 0.79) and HGBC (AUC 0.78) for 1-year mortality. Age and NT-proBNP were the strongest predictors in SHAP analyses for both outcomes.

## Conclusion

Models derived from EHR data can predict mortality after HFpEF hospitalization with comparable performance to models derived from registry or trial data, highlighting the potential for clinical implementation.

---

## Introduction

Heart failure with preserved ejection fraction (HFpEF) is a distinct subtype of heart failure (HF), and accounts for the majority of HF hospitalizations [1]. Despite this burden of hospitalizations, and the associated considerable morbidity and mortality, prognostic models specifically for patients hospitalized with HFpEF are less established. Accurate prediction models are essential to physicians to help identify and manage high risk patients, to health systems for allocating resources, and to policy makers for risk adjustment to measure performance.

With the wide availability of electronic health records (EHR), there is a need for predictive models to be based on real-world EHR data which is critical for implementation at the bedside. The few predictive models that have been developed for HFpEF have been derived from registry or trial data and are for ambulatory populations [2–7]. In addition, these models often contain variables such as New York Heart Association (NYHA) Class or complex health status assessments, which are not readily available in the EHR [3–6]. Additionally, it is important for models to be developed in a data-driven approach incorporating complex interactions, which can be accomplished with machine learning techniques.

Accordingly, we leveraged data from the Medical Information Mart for Intensive Care (MIMIC)-IV database and tested a variety of modeling techniques including machine learning to develop prediction models for 30-day and 1-year mortality with an index hospitalization for HFpEF. We compared model performance using an array of performance metrics.

## Methods

This study adheres to the guidelines set by the Transparent Reporting of a multivariable prediction model for Individual Prognosis Or Diagnosis (TRIPOD) statement. Compliance with the TRIPOD checklist for the thorough and transparent reporting of our predictive model development and validation processes are detailed in S1 Table [8].

We employed seven predictive models: Logistic Regression, Lasso Regression, Elastic Net, Support Vector Classifier (SVC) with a radial basis function (RBF) kernel,

Random Forest, Histogram-based Gradient Boosting Classifier (HGBC), and eXtreme Gradient Boosting (XGBoost) [9–15]. Each model class has its unique advantages in handling different aspects of the data.

The models were evaluated using the following metrics: Accuracy, Sensitivity, Specificity, Area Under the ROC curve (AUC), Precision-Recall Area Under the Curve (PR-AUC), Calibration curves, MCC score (Matthews Correlation Coefficient), AIC (Akaike Information Criterion), and BIC (Bayesian Information Criterion) [16–19].

Accuracy, Sensitivity, Specificity, AUC and PR-AUC are commonly encountered metrics used to evaluate models in medical literature. In addition, MCC is a balanced measure of model performance, particularly in the context of imbalanced classes, as it considers true and false positives and negatives, offering more information than accuracy alone. AIC and BIC both assess model fit and complexity. AIC estimates the relative quality of models for a given dataset by considering the trade-off between goodness-of- fit and the number of parameters, penalizing models with excessive complexity. BIC incorporates a penalty term for the number of parameters but with a stronger penalty for model complexity, providing a stricter criterion that favors more parsimonious models. We informed overall model selection with the metrics that would be more important from a clinical standpoint for this particular prediction problem.

## Data sources

We used the MIMIC-IV dataset version 2.2 – a publicly shared database of de- identified electronic health record data, including hospital and intensive care unit admissions from the Beth Israel Deaconess Medical Center in Boston, MA from 2008 to 2019 [20,21]. The data were accessed via PhysioNet after completing the necessary requirements. Patients and/or the public were not involved in the design, or conduct, or reporting, or dissemination plans of this research. The data that support the findings of this study are openly available in Physionet [22]. Given that the MIMIC IV data is de-identified and publicly accessible, the study was not subject to Yale Institutional Review Board review.

## Study population

We identified hospitalizations from 2008 to 2019 of patients aged ≥ 18 years with HFpEF as a primary diagnosis using appropriate ICD-9 and ICD-10 codes (S2 Table) [23].

As our diagnosis was based on ICD codes, to test the validity of this label we queried clinical notes using regular expressions to extract mentions of the left ventricular ejection fraction value or a qualitative report of the left ventricular function using appropriate phrases. However, as this LVEF data was extracted from clinical notes and not readily available in a structured field in MIMIC-IV data, we intentionally did not include this in predictive modeling.

From a total of 430,852 hospitalizations, we identified 3,235 individual hospitalization encounters with a discharge diagnosis of HFpEF which comprised the study sample. Among these hospitalization encounters, we had access to clinical notes for 3,146 (97.3%) encounters of which 1,836 (58.4%) had an LVEF measurement value reported. Of these, 1,726 (94.0%) had an LVEF value ≥ 50%, and 46 (2.5%) had an LVEF between 45-50%. An additional 586 (18.6%) encounters had a qualitative mention of LVEF, of which 551 (94.0%) indicated the LVEF was normal/preserved. Thus, this ICD code-based diagnosis label was considered valid for identifying encounters with HFpEF within MIMIC-IV data. Given that among the 77% encounters with either a quantitative or qualitative mention of ventricular mention, >97% had a documented preserved LVEF we concluded that this method of ICD code-based identification of HFpEF is valid and has sufficient positive predictive value. This validation exercise will also be useful for future projects by other researchers that use ICD codes to identify HFpEF. In this process we did note 55 cases with a documented LVEF <40% and 16 cases with a qualitative mention of 'reduced' LVEF. However, these cases were not excluded from the cohort as the primary method of identification for this study remained ICD code-based. The notes assessment was primarily for the purpose of validation of the ICD code-based diagnosis.

## Outcomes

Outcomes for predictive models included 30-day and 1-year mortality. Date of death in MIMIC data is derived from hospital records and state records. The maximum time of follow up for each patient in MIMIC data is exactly one year after their last hospital discharge.

## Data extraction

Data containing patient demographics, vital signs, diagnoses using ICD codes, admission information, laboratory tests, and date of death were extracted from appropriate relational tables using two identification columns: 'subject_id' and 'hadm_id'. The 'subject_id' represents a single patient's admission to the hospital, while the 'hadm_id' pertains to a specific hospital admission event. For data tables not readily alignable through these IDs, we employed alternative matching strategies, such as correlating timestamps within one day.

ICD diagnosis codes were mapped to comorbidity categories in the Charlson Comorbidity Index (CCI) - a common method for mapping and summarizing patients' comorbidities. However, as the weighting of comorbidities in CCI is not particular to HFpEF, and our goal is to identify and use predictive variables, we did not use the comorbidity score as a predictive variable and instead used the individual mapped comorbidities as separate variables. In addition, we included a select few other comorbidities such as hypertension, atrial fibrillation, pulmonary hypertension etc. which are noted to be predictors in prior HFpEF prediction models but are not a part of the CCI comorbidities. For vital signs and specific lab values we used the first entry on the day of admission using appropriate time stamps.

We assessed sample size adequacy to support model development to predict mortality in HFpEF patients by using the criteria suggested by Riley et al [24]. Using the I- PRESERVE[4] 1-year all-cause mortality model's AUC of 0.74 as a benchmark, we calculated the minimum sample size required for 1-year mortality with a prevalence of 29.2% to be 2,037. For 30-day mortality there are no contemporary prediction models for HFpEF specific to this time frame. However, using an in-hospital mortality model by Wang et al. with an AUC of 0.83 as a reference, and an observed 30-day mortality rate of 6.3% in our cohort, a similarly performing model would need a sample size of 3,186 [25]. This suggested our sample size should be adequate for both outcomes.

## Preprocessing

As a part of data preprocessing, we one-hot encoded sex, and binarized comorbidity variables. Four extreme outliers were identified and subsequently treated as missing data. Based on visual inspection of the variable distributions, we applied PowerTransformer (Yeo–Johnson) or QuantileTransformer (normal output) transformations to variables with wide ranges or evident non-normal distributions (e.g., creatinine, INR, platelet count, WBC count, and oxygen saturation), while BMI was standardized to zero mean and unit variance. These transformations were applied prior to training Elastic Net, Lasso, Logistic Regression, SVC, and XGBoost models. Random Forest and HGBC models were trained on the original unscaled features, as tree-based methods are inherently robust to feature distribution and scaling.

To identify the most effective preprocessing strategy for handling missing data, we explored several imputation techniques, including mean and median imputation, along with Multiple Imputation by Chained Equations (MICE). As a validation, we compared the statistical analyses results from the imputed data with those obtained after dropping missing data and assessed the consistency of results and distribution changes to best maintain data integrity and statistical power, while avoiding the substantial data loss associated with dropping missing data. The statistical tests included the Shapiro-Wilk test for normality, t-tests, and Mann-Whitney U tests for continuous variables, Chi-Squared and Fisher's Exact tests for categorical variables, and Variance Inflation Factor (VIF) analysis for multicollinearity.

To address class imbalance, we employed random oversampling, undersampling, Synthetic Minority Over-sampling Technique (SMOTE), and balanced sampling methods. Each method was evaluated based on final performance metrics

to determine its effectiveness in creating a balanced class distribution and improving the performance and generalizability of our predictive models, with model performance also assessed using a baseline of no imputation and no resampling for comparison.

Imputation was done first, followed by transformations, resampling, and then scaling (for applicable models).

### Feature analysis

Our study included a set of 36 features selected based on data availability and clinical relevance – 17 categorical and 19 continuous features (S3 Table). Categorical features included patient demographics, and comorbidities (as defined by ICD codes) such as diabetes, renal disease, and cancer. Continuous features included vital signs such as heart rate, systolic blood pressure, and oxygen saturation, laboratory values like hemoglobin, creatinine, sodium troponin and NT-proBNP levels. Note that prior renal disease as identified by ICD codes and admission Creatinine values were both used as individual variables in model development.

To understand the relationship between individual features and their predictive power, mutual information plots for 30-day and 1-year mortality were constructed. Additionally, Pearson correlation heatmaps were generated to visualize the linear relationships between continuous features.

### Model fitting and evaluation

An 80−20 data split was applied to separate the data into training and testing sets. We used 5-fold cross-validation for the pipelines utilizing resampling methods, and for the pipeline without resampling methods, we utilized a repeated stratified K-Fold cross- validation, considering its strength towards the imbalance classification task. We used a randomized hyper-parameter search to fine-tune each model. Model evaluation was performed using the metrics outlined above.

### Model interpretability and explainability

To enhance the transparency and interpretability of our predictive models, we used SHAP (SHapley Additive exPlanations) values which provide a unified measure of feature importance, quantifying the contribution of each feature to the model's predictions. We used SHAP summary plots and bar plots to visualize the global importance of features. For logistic regression models, we calculated odds ratios to quantify the impact of each feature on the target variable.

Analyses were conducted using Python 3.10.12, R, and Stata Statistical Software: Release 18 (College Station, TX). Code used to analyze data and build the models is publicly accessible via [GitHub](https://github.com/itsjustnilay/Predictive_Modelling_for_Heart_Failure_with_Preserved_Ejection_Fraction).

### Results

The study sample consisted of 3,235 individual hospitalization encounters with a discharge diagnosis or HFpEF. Demographics and clinical characteristics for the study sample are shown in Table 1. Note that the comorbidities in Table 1 were defined by ICD codes. The mean age of the study population was 76.4 ± 13.3; 62.0% were female, and 20.5% self-identified as Black. Missing values proportions by variable are shown in S4 Table. BMI, temperature, and oxygen saturation had higher proportions of missing values, while laboratory parameters like Creatinine, Bicarbonate, and Hemoglobin had fewer missing values, except for troponin which had a high proportion of missing.

The observed 30-day mortality was 6.3% (N=245) and 1-year mortality was 29.2% (N = 1145). Women had similar mortality to men (28.5% vs 27.5%, p=0.52). The in- hospital mortality rate for Black patients was lower at 20.7% vs 31.6% for White, while that for patients ≥65 years was higher at 31.6% vs 13.9% for those <65 years (both p<0.001). Patients who died during their hospital stay had higher proportions of comorbidities such as chronic kidney disease, chronic obstructive pulmonary disease (COPD), cancer, atrial fibrillation, compared with patients who survived hospitalization (Table 1).

**Table 1. Baseline Characteristics of Patients by Survival Status (N = 3,235).**

| | 30-Day | | 1-Year | |
| --- | --- | --- | --- | --- |
| Outcome | Survived (n = 3,051) | Death (n = 184) | Survived (n = 2,325) | Death (n = 910) |
| **Demographics** | | | | |
| Age, years (mean ± std)* | 76.03 ± 13.34 | 83.35 ± 9.6 | 74.49 ± 13.46 | 81.45 ± 11.3 |
| Race, n (%)* | | | | |
| White | 2040 (66.86) | 159 (86.41) | 1505 (64.73) | 694 (76.26) |
| Hispanic | 130 (4.26) | 3 (1.63) | 111 (4.77) | 22 (2.42) |
| Black | 648 (21.24) | 15 (8.15) | 526 (22.62) | 137 (15.05) |
| Asian | 102 (3.34) | 2 (1.09) | 74 (3.18) | 30 (3.30) |
| Others | 131 (4.29) | 5 (2.72) | 109 (4.69) | 27 (2.97) |
| Sex, n (%) | | | | |
| Female | 1893 (62.05) | 112 (60.87) | 1433 (61.63) | 572 (62.86) |
| **Vital signs (mean ± std)** | | | | |
| Temperature, °F* | 98.08 ± 0.87 | 97.85 ± 0.91 | 98.11 ± 0.86 | 97.94 ± 0.92 |
| Heart rate, bpm | 79.94 ± 17.47 | 82.65 ± 17.56 | 79.69 ± 17.61 | 81.33 ± 16.94 |
| Oxygen saturation, % | 96.69 ± 3.75 | 96.19 ± 5.23 | 96.65 ± 3.84 | 96.73 ± 3.79 |
| Systolic BP, mmHg* | 138.4 ± 25.30 | 127.52 ± 23.12 | 139.5 ± 25.85 | 132.64 ± 22.57 |
| BMI, kg/m²* | 33.52 ± 11.01 | 28.56 ± 6.38 | 34.43 ± 11.16 | 29.81 ± 9.17 |
| **Lab values (mean ± std)** | | | | |
| Bicarbonate, mmol/L | 28.07 ± 4.69 | 27.77 ± 5.45 | 28.01 ± 4.55 | 28.17 ± 5.17 |
| Creatinine, mg/dL | 1.76 ± 1.49 | 1.72 ± 0.96 | 1.71 ± 1.51* | 1.88 ± 1.34* |
| Hemoglobin, g/dL | 10.50 ± 1.90 | 10.47 ± 1.75 | 10.61 ± 1.92* | 10.21 ± 1.79* |
| INR | 1.83 ± 0.92* | 2.05 ± 1.13* | 1.83 ± 0.92 | 1.89 ± 0.97 |
| Platelet count, 10³/µL | 232.03 ± 93.64 | 233.34 ± 110.33 | 233.76 ± 91.33 | 227.90 ± 102.53 |
| Potassium, mmol/L* | 4.09 ± 0.55 | 4.20 ± 0.66 | 4.08 ± 0.55 | 4.13 ± 0.58 |
| WBC count, 10³/µL* | 7.91 ± 4.87 | 10.53 ± 12.54 | 7.85 ± 4.57 | 8.61 ± 7.66 |
| Sodium, mmol/L* | 139.12 ± 4.20 | 137.89 ± 5.10 | 139.16 ± 4.11 | 138.79 ± 4.63 |
| NT-proBNP, pg/mL* | 6178.22 ± 8837.22 | 11794.53 ± 11729.97 | 5269.31 ± 7951.15 | 9719.04 ± 11003.36 |
| Troponin, ng/mL | 0.11 ± 0.45 | 0.18 ± 0.39 | 0.11 ± 0.53 | 0.12 ± 0.24 |
| **Comorbidities, n (%)** | | | | |
| Peripheral vascular disease* | 318 (10.42) | 30 (16.30) | 234 (10.06) | 114 (12.53) |
| Cerebrovascular disease | 178 (5.83) | 14 (7.61) | 122 (5.25)* | 70 (7.69)* |
| Chronic obstructive pulmonary disease | 1445 (47.36) | 93 (50.54) | 1057 (45.46)* | 481 (52.86)* |
| Rheumatoid disease | 154 (5.05) | 7 (3.80) | 114 (4.90) | 47 (5.16) |
| Peptic ulcer disease | 37 (1.21) | 0 (0.00) | 31 (1.33) | 6 (0.66) |
| Mild liver disease | 152 (4.98) | 10 (5.43) | 107 (4.60) | 55 (6.04) |
| Renal disease | 1516 (49.69) | 99 (53.80) | 1095 (47.10)* | 520 (57.14)* |
| Moderate severe liver disease | 33 (1.08) | 4 (2.17) | 18 (0.77)* | 19 (2.09)* |
| Acute myocardial infarction | 403 (13.21) | 29 (15.76) | 295 (12.69) | 137 (15.05) |
| Dementia | 104 (3.41) | 10 (5.43) | 69 (2.97)* | 45 (4.95)* |
| Diabetes | 1019 (33.40) | 49 (26.63) | 793 (34.11)* | 275 (30.22)* |
| Diabetes complications | 499 (16.36) | 21 (11.41) | 404 (17.38)* | 116 (12.75)* |
| Hemiplegia paraplegia | 7 (0.23) | 1 (0.54) | 5 (0.22) | 3 (0.33) |
| Cancer* | 201 (6.59) | 24 (13.04) | 125 (5.38) | 100 (10.99) |
| Metastatic cancer* | 51 (1.67) | 13 (7.07) | 22 (0.95) | 42 (4.62) |
| Hypertension* | 1374 (45.03) | 64 (34.78) | 1108 (47.66) | 330 (36.26) |

*(Continued)*

**Table 1.** (Continued)

|  | 30-Day |  | 1-Year |  |
| --- | --- | --- | --- | --- |
| Coronary artery disease | 1197 (39.23) | 75 (40.76) | 894 (38.45) | 378 (41.54) |
| Pulmonary hypertension | 858 (28.12) | 55 (29.89) | 625 (26.88)* | 288 (31.65)* |
| Atrial fibrillation* | 1526 (50.02) | 126 (68.48) | 1074 (46.19) | 578 (63.52) |
| **Discharge Medications, n (%)†** |  |  |  |  |
| Ace Inhibitors | 841 (27.59) | 20 (16.81) | 683 (29.38) | 178 (21.14) |
| Angiotensin II receptor blockers (ARBs) | 424 (13.91) | 9 (7.56) | 366 (15.74) | 67 (7.96) |
| Beta blockers | 2184 (71.58) | 74 (40.22) | 1689 (72.65) | 569 (67.58) |
| Diuretics | 2240 (73.49) | 66 (55.46) | 1729 (74.37) | 577 (68.53) |
| Aspirin | 1863 (61.12) | 61 (51.26) | 1419 (61.03) | 505 (59.98) |
| P2Y12 inhibitors | 260 (8.53) | 7 (5.88) | 207 (8.90) | 60 (7.13) |
| Aldosterone Antagonists | 305 (10.01) | 13 (10.92) | 230 (9.89) | 88 (10.45) |
| Anticoagulants | 1136 (37.27) | 39 (32.77) | 868 (37.33) | 307 (36.46) |

*Shows the statistical significance at the α = 0.05 level; † Discharge medications are reported only for patients who survived to hospital discharge.

Correlation heat maps for continuous variables are shown in S1 Fig. and Mutual information plots are shown in S2 Fig. Mutual information plots showed NT-proBNP and age to be key predictors for both outcomes, while heart rate, White race, and potassium levels were significant markers for 30-day mortality, while systolic blood pressure, Black race, and oxygen saturation were significant predictors for one-year mortality. Black race has been previously shown to be associated with lower mortality in HFpEF [26]. However, as race is a social and not a biological construct, we did not include any race variables in predictive modeling. Multiple imputation and balanced resampling methods were noted to be the most effective strategies for managing missing and class imbalance respectively.

Model performance metrics are shown in Table 2. and AUC curves for all models are shown in Fig 1. PR-AUC and calibration curves are shown in S3 and S4 Figs. respectively.

**Table 2.** Performance Metrics of Predictive Models for 30-Day and 1-Year Mortality.

| Outcome | Model | Imputation | Resampling | Accuracy | MCC | Sensitivity | Specificity | AIC | BIC | PR-AUC | AUC |
| --- | --- | --- | --- | --- | --- | --- | --- | --- | --- | --- | --- |
| 30-day | LR[a] | Median | Undersampling | 0.67 | 0.23 | 0.82 | 0.66 | 859.09 | 1024.57 | 0.33 | 0.83 |
|  | Lasso | Mean | Undersampling | 0.71 | 0.19 | 0.74 | 0.65 | 921.16 | 1086.64 | 0.31 | 0.82 |
|  | Elastic Net | Median | Undersampling | 0.66 | 0.19 | 0.74 | 0.66 | 923.78 | 1089.26 | 0.30 | 0.82 |
|  | SVC[b] | Mean | Undersampling | 0.61 | 0.17 | 0.76 | 0.60 | 891.24 | 1056.72 | 0.18 | 0.75 |
|  | RF[c] | Median | Undersampling | 0.69 | 0.20 | 0.71 | 0.68 | 814.10 | 979.58 | 0.18 | 0.78 |
|  | HGBC[d] | Multiple | Undersampling | 0.68 | 0.22 | 0.76 | 0.68 | 2447.41 | 2612.88 | 0.23 | 0.75 |
|  | XGBoost[e] | Mean | Undersampling | 0.70 | 0.20 | 0.71 | 0.70 | 1009.57 | 1175.04 | 0.19 | 0.75 |
| 1-year | LR | Median | None | 0.78 | 0.36 | 0.36 | 0.93 | 722.82 | 888.30 | 0.57 | 0.75 |
|  | Lasso | Median | None | 0.79 | 0.39 | 0.34 | 0.95 | 725.29 | 890.76 | 0.57 | 0.74 |
|  | Elastic Net | Median | None | 0.79 | 0.38 | 0.35 | 0.94 | 723.80 | 889.28 | 0.57 | 0.75 |
|  | SVC | Median | Undersampling | 0.67 | 0.34 | 0.72 | 0.66 | 836.28 | 1001.76 | 0.53 | 0.75 |
|  | RF | Multiple | None | 0.77 | 0.32 | 0.26 | 0.96 | 698.25 | 863.73 | 0.59 | 0.79 |
|  | HGBC | Multiple | Oversampling | 0.77 | 0.38 | 0.49 | 0.87 | 1082.89 | 1248.36 | 0.61 | 0.78 |
|  | XGBoost | Multiple | Oversampling | 0.78 | 0.39 | 0.47 | 0.89 | 948.69 | 1114.17 | 0.60 | 0.77 |

[a]LR, Logistic Regression; [b]SVC, Support Vector Classifier; [c]RF, Random Forest; [d]HGBC, Histogram-based Gradient Boosting Classifier; [e]XGBoost, eXtreme Gradient Boosting.

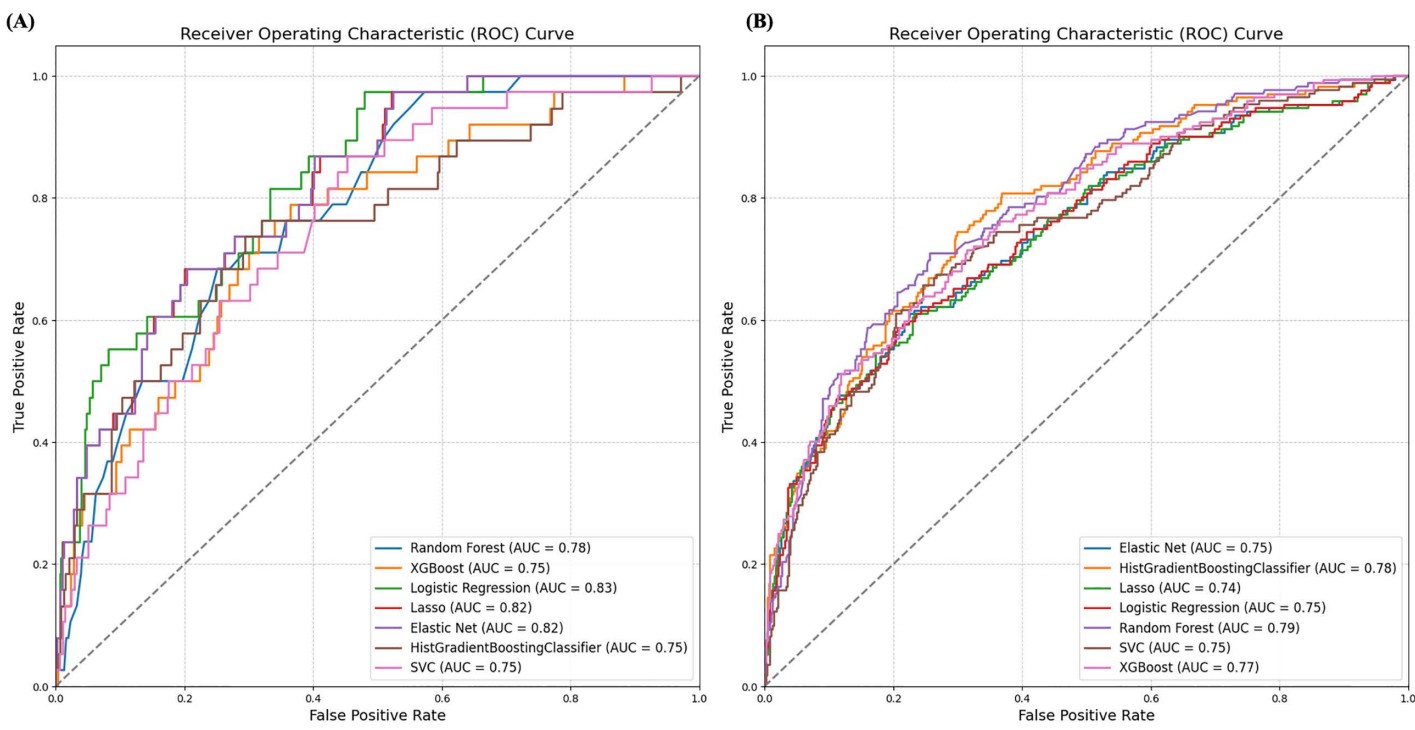

**Fig 1. ROC Curves for Predictive Models of (A) 30-Day Mortality and (B) 1-Year Mortality.**

## Model performance

For 30-day mortality, the regression-based models overall appeared to perform better than tree-based models. The Logistic Regression model using median imputation and random under-sampling demonstrated an overall good performance with an accuracy of 0.67, AUC of 0.83, sensitivity of 0.82, and specificity of 0.66.

For 1-year mortality, tree-based models overall appear to perform better. The HGBC model using multiple imputation and random oversampling had an accuracy of 0.77, AUC of 0.78, sensitivity of 0.49, and specificity of 0.87. On the other hand, regression models such as Elastic Net model showed higher specificity but lower sensitivity (accuracy of 0.79, AUC of 0.75, sensitivity of 0.35, and specificity of 0.94).

## Variable importance

The odds ratios (OR) for the logistic regression models for 1-year and 30-day mortality are shown in Table 3. For 30-day mortality, the most significant predictors were elevated WBC count and NT-proBNP levels (OR: 2.85 and 2.44 respectively). Other important predictors included age, troponin and bicarbonate levels. For 1-year mortality, age and elevated NT-proBNP levels (Odds Ratio: 1.78 and 1.66 respectively) were significant predictors, though with lower odds ratios compared to 30-day mortality. Atrial fibrillation, metastatic cancer, and elevated bicarbonate level were other important predictors.

## Interpretability and explainability

SHAP summary plots for 30-day and 1-year mortality are shown in Fig 2 and SHAP bar plots in S5 Fig. SHAP interpretations were performed for the Logistic regression model for 30-day mortality outcome and HGBC for 1-year mortality. For

**Table 3. Feature Odds Ratios for 30-Day and 1-Year Mortality as per Logistic Regression.**

| 30-Day Mortality | | 1-Year Mortality | |
|---|---|---|---|
| Feature | Odds Ratio | Feature | Odds Ratio |
| WBC count | 2.846 | Age | 1.780 |
| NT-proBNP | 2.444 | NT-proBNP | 1.658 |
| Age | 2.305 | Atrial fibrillation | 1.324 |
| Troponin | 2.000 | Metastatic cancer | 1.273 |
| Bicarbonate | 1.492 | Bicarbonate | 1.250 |
| Peripheral vascular disease | 1.401 | Chronic obstructive pulmonary disease | 1.234 |
| Potassium | 1.400 | Moderate/severe liver disease | 1.203 |
| Metastatic cancer | 1.300 | WBC count | 1.190 |
| Atrial fibrillation | 1.293 | Heart rate | 1.123 |
| Moderate severe liver disease | 1.275 | Cancer | 1.108 |

30-day mortality, NT-proBNP was the most important feature, followed by age and coronary artery disease. For 1-year mortality, age at admission, NT-proBNP levels and systolic blood pressure levels were the most significant factors.

## Discussion

In our study, models derived from EHR data to predict 30-day and 1-year mortality with a Heart Failure with Preserved Ejection Fraction (HFpEF) hospitalization showed good performance and potential for clinical use. Regression models performed well for the 30-day outcome with the overall best performing Logistic regression model with an AUC of 0.83. Tree-based models overall appear to perform better for the 1-year outcomes with the best performing HGBC model with an AUC of 0.78.

Prior studies developing prediction models in HFpEF have focused on the ambulatory population [4,6,7,27] and are not optimal to be used in the hospitalized setting, where markers of acuity such as vital signs etc. need to be additionally incorporated and can help define risk. Further, most prior HFpEF models have been derived from trial data which have standardized data collection, and often contain variables which are not readily available in the EHR, such as complex health status assessment, NYHA Class, or genetic data. Additionally, traditional models often focus on being parsimonious, which is extremely pertinent for low resource settings, but in clinical environments delivering care using contemporary EHR systems, computation is not a limitation, and thus leveraging all available variables and modeling the complexity of variable relationships can help improve risk prediction [28].

It is critical for models to be developed using EHR data for two reasons. First, patient populations sourced from the EHR may be more reflective of the real-world than trial data which can be affected by selection bias. Second, EHR-based prediction models are easier to implement in patient facing environments, given that the constituent risk variables are already sourced from EHR and are obtained in routine clinical care. The potential uses of these models could be early risk stratification for in-hospital planning such as ICU triage, and triggering care pathways like advanced HF team involvement, palliative care involvement etc. There is also a role for quality metrics and for hospitals to track post-discharge outcomes.

In our study, for predicting 30-day mortality, regression-based models (Logistic regression and Elastic Net) performed better than tree-based models. The logistic regression model had the best metrics overall including an AUC of 0.83. It could be that short-term outcomes are driven by more immediate and linear relationships with acute clinical indicators which are modeled well by regression methods. Additionally, it may be that regression-based methods are able to handle the highly imbalanced nature of the 30-day outcome more effectively. Techniques like Elastic Net provide regularization, preventing overfitting by penalizing complex models, which may be crucial for the shorter prediction window. In addition,

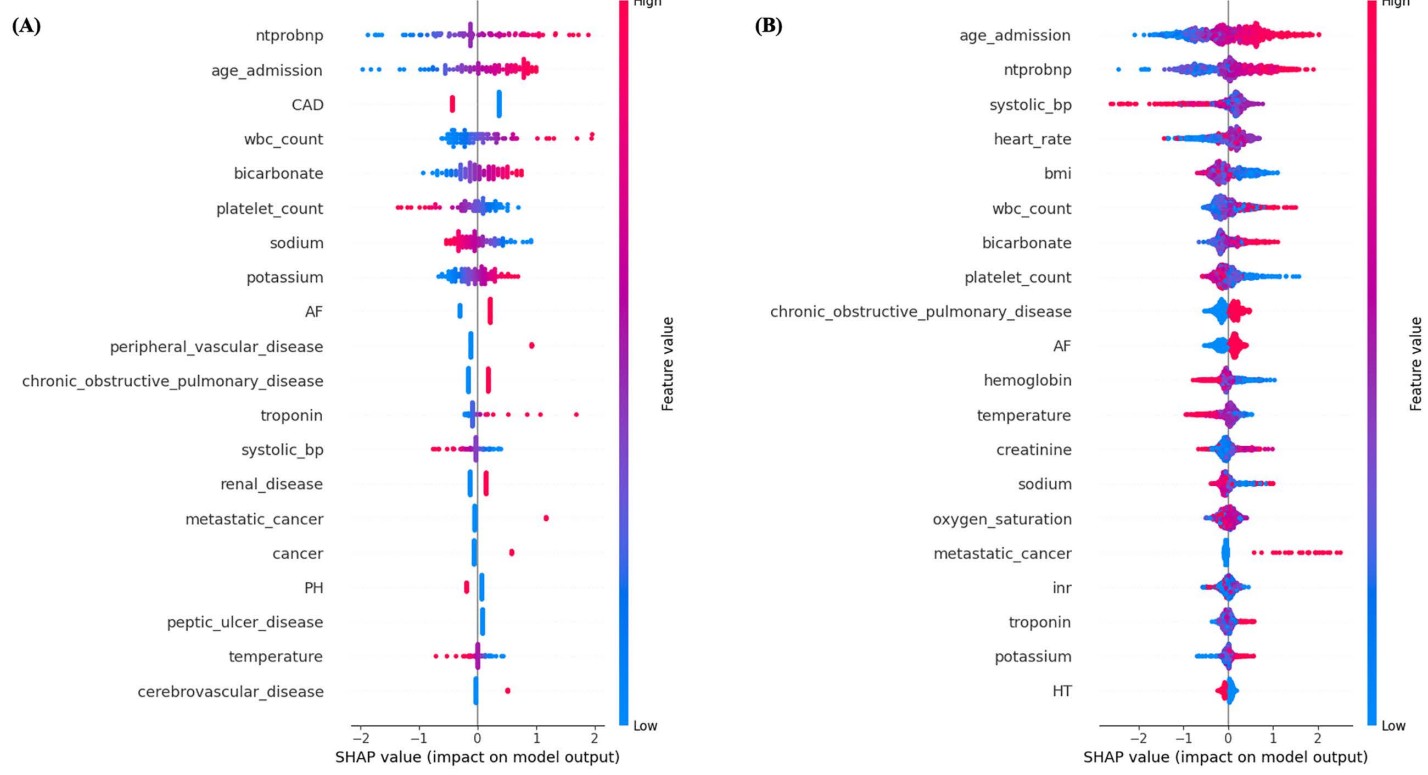

**Fig 2. SHAP Summary Plots for Predictive Models of (A) 30-Day Mortality and (B) 1-Year Mortality.**

as short-term mortality risk is often incorporated into triage decisions, the higher sensitivity of the regression-based models is also favorable.

Tree-based models, on the other hand, performed better for the 1-year outcome with the overall best performing HGBC model with an AUC of 0.78. Tree-based models are non-linear, which enables them to capture complex interactions between variables which are present in long-term prediction tasks. They are also more effective at handling different types of data and missing values, ensuring robust prediction in the face of incomplete data. Their ensemble-based structure aggregating predictions from multiple trees, makes them versatile and helps reduce variance. They also improve predictive accuracy by leveraging the strengths of multiple models to explore deeper relationships within the data, capturing long-term trends and patterns more effectively than linear models.

Age was an important predictor of both 30-day and 1-year mortality. Sex was not a predictor unlike noted in some other prior models [27]. Among comorbidities, we noted COPD, atrial fibrillation, malignancy and liver disease to be important predictors. Among laboratory parameters NT-pro-BNP was the most important predictor, as has been noted in most prior models in HFpEF, and affected both outcomes significantly [4,25,27,28]. Troponin on the other hand was an important predictor more for 30-day mortality than for 1 year. A higher bicarbonate level and wbc count, similar to a prior study, were also noted to be an important variable for both outcomes. Unlike in HFrEF, the effect of elevated bicarbonate levels on mortality in HFpEF have not been specifically reported before [4,29].

Overall, in terms of the individual risk variables most are not novel such as age and NT-proBNP and have been noted in prior studies. However, the novelty of our study remains the use of real-world EHR data from a large health system to derive the model and the being model specific to the hospitalized population. In contrast prior predictive models for HFpEF

have been derived from registry or trial data and are for ambulatory populations. Thus, although there is some commonality of variables identified with prior models, the strength of association with the outcomes may differ in this treatment setting. Further, the incomplete nature of EHR data also can change the association of variables. An EHR derived model is likely to perform better in a patient-facing EHR setting. However, despite these advantages further validation of our model is needed.

To further enhance the predictive accuracy of such EHR-based models, future investigations could use data combined from multiple health systems, which will allow larger numbers of patients, to fully leverage the capabilities of machine learning methodologies. In addition, exploring ensemble methods by combining model classes can further enhance prediction by strategically amalgamating the strengths of individual algorithms. Further, including additional data categories such as prescription fill data and imaging parameters can help enhance prediction. These data streams are currently not universally accessible in EHRs, however, with advancements in interoperability there is a potential in the near future for incorporating such data and more into clinical models for use at the bedside.

## Limitations

One limitation of our study is the lack of external validation using an independent cohort, and despite the use of techniques like stratified cross-validation and bootstrapping concerns remain of the model's generalizability. Further validation across diverse populations is necessary. Additionally, the completeness of the data presented challenges, particularly with features that exhibited high levels of imbalance and missingness. This is however, a common issue encountered with EHR data. Although imputation and resampling methods were carefully applied to address these issues to maintain the original dataset's distribution, these processes can introduce bias and leave the potential for misclassification, which may impact the model's performance. Despite implementing regularization techniques to reduce the risk of overfitting, there remains a concern that the model may still be overly tailored to the training data.

## Conclusion

Models derived from EHR data have good performance in predicting 30-day and 1-year mortality with a HFpEF hospitalization, with performance metrics similar to other contemporary models derived from trial datasets. Models derived from EHR have an immediate potential to be implemented at the bedside.

## Supporting information

**S1 Table. Transparent Reporting of a multivariable prediction model for Individual Prognosis Or Diagnosis (TRIPOD) statement.**
(PDF)

**S2 Table. ICD codes of observed outcomes and their frequencies.**
(PDF)

**S3 Table. Features, Target Outcomes and their data types.**
(PDF)

**S4 Table. Missing Values (count and %) of features.**
(PDF)

**S1 Fig. Correlation Matrix for Continuous Variables in (A) 30-Day and (B) 1-Year Mortality.**
(PDF)

**S2 Fig. The mutual information (MI) analysis comparing (A) 30-day and (B) 1-year mortality.**
(PDF)

**S3 Fig. The precision-recall curves for (a) 30-Day and (b) 1-Year mortality outcomes.**
(PDF)

**S4 Fig. The calibration curves for (A) 30-Day and (B) 1-Year mortality outcomes.**
(PDF)

**S5 Fig. SHAP Bar plots for (A) 30-Day mortality Logistic regression model and (B) 1-Year mortality outcomes HGBC model.**
(PDF)

## Author contributions

**Conceptualization:** Alaa Alashi, Karthik Murugiah.

**Data curation:** Ikgyu Shin, Alaa Alashi, Karthik Murugiah.

**Formal analysis:** Ikgyu Shin, Nilay Bhatt, Alaa Alashi, Karthik Murugiah.

**Funding acquisition:** Karthik Murugiah.

**Investigation:** Ikgyu Shin, Alaa Alashi, Keervani Kandala, Karthik Murugiah.

**Methodology:** Ikgyu Shin, Nilay Bhatt.

**Project administration:** Ikgyu Shin.

**Resources:** Ikgyu Shin, Nilay Bhatt, Alaa Alashi.

**Software:** Ikgyu Shin, Nilay Bhatt, Alaa Alashi.

**Supervision:** Ikgyu Shin, Alaa Alashi, Karthik Murugiah.

**Validation:** Ikgyu Shin, Karthik Murugiah.

**Visualization:** Ikgyu Shin, Nilay Bhatt.

**Writing – original draft:** Ikgyu Shin, Karthik Murugiah.

**Writing – review & editing:** Ikgyu Shin, Nilay Bhatt, Alaa Alashi, Keervani Kandala, Karthik Murugiah.

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
