## [Decision Letter · Decision Letter 0]

9 Jul 2025

Predicting 30-day and 1-year mortality in heart failure with preserved ejection fraction (HFpEF)

PLOS ONE

Dear Dr. Murugiah,

Thank you for submitting your manuscript to PLOS ONE. After careful consideration, we feel that it has merit but does not fully meet PLOS ONE’s publication criteria as it currently stands. Therefore, we invite you to submit a revised version of the manuscript that addresses the points raised during the review process.

We look forward to receiving your revised manuscript.

Kind regards,

Shukri AlSaif

Academic Editor

PLOS ONE

Journal Requirements:

 “Dr. Murugiah received support from the National Heart, Lung, and Blood Institute of the National Institutes of Health (under award K08HL157727).” 

Please state what role the funders took in the study.  If the funders had no role, please state: 'The funders had no role in study design, data collection and analysis, decision to publish, or preparation of the manuscript.'

Reviewers' comments:

Reviewer's Responses to Questions

**Comments to the Author**

1. Is the manuscript technically sound, and do the data support the conclusions?

Reviewer #1: Yes

2. Has the statistical analysis been performed appropriately and rigorously?

Reviewer #1: Yes

3. Have the authors made all data underlying the findings in their manuscript fully available?

Reviewer #1: Yes

4. Is the manuscript presented in an intelligible fashion and written in standard English?

Reviewer #1: Yes

Reviewer #1: Summary:

This study investigates prognostic models of 30-days and 1-year mortality in HFpEF patients using EHR data sources. The authors demonstrate that logistic regression model predicted 30-day mortality

whereas Random Forest and Histogram-based Gradient Boosting Classifier predicted 1-year mortality. Age and NT-proBNP were the strongest predictors in SHapley Additive exPlanations analyses for both outcomes.

Overall, the study design is adequate, and the paper is clearly written. The presented results seem valid and reliable. The analyses are performed in real-world data.

Major comment and Clinical utility:

Rigorously performed analyses demonstrating the use of advanced models. These can be used as algorithms assisting clinicians if based on few clinically available variables and highlighting certain variables of importance. The present data shows the importance of age and NPs as prognostic predictors. Can be questioned if this information adds to the clinicians’ decision and risk assessment beyond what we already know.

Specific points:

Abstract:

-

Background:

Misspelling at row 68 (HER).

Methods:

Regarding the selection of patients based on ICD codes. Were patients with recorded LVEF =<40% in stable condition or during a later hospitalization excluded?

Was data extracted for the full time period 2008-2019?

For vital signs and specific lab values first entry on the day of admission were used. In a clinical perspective using variables in a clinical model the last captured measurements before discharge may be more relevant.

Logarithmic transformations were used for certain “variables with wide ranges”. Also for variables with non-normal distribution?

Line 173 states gender was one-hot encoded – do authors refer to sex (binary)? Gender consists of >25 different categories.

Consider including eGFR in models instead of creatinine

Results:

Table 1

How was renal disease defined? ICD codes or per eGFR?

Treatment should be of interest to include in Table 1, even if not included in models.

Fig 1, 2 and S2

Are in low solution and hard to read due to the small font.

Discussion:

A discussion on how this kind of models can be used in clinical practice would add to the manuscript.

**Do you want your identity to be public for this peer review?** For information about this choice, including consent withdrawal, please see our Privacy Policy

Reviewer #1: **Yes: ** Camilla Hage

---

## [Author Response · Author response to Decision Letter 1]

25 Aug 2025

Dear Dr. Chenette and Editor Dr. Hage,

Thank you for the opportunity to revise and resubmit our manuscript titled “Predicting 30-Day and 1-Year Mortality in Heart Failure with Preserved Ejection Fraction (HFpEF)” to Plos One. We thank the reviewers of Plos One for their detailed comments on the manuscript and have edited the manuscript to address their concerns, which has improved the research paper. The line-by-line responses to the reviewer comments are provided in the document Response to Reviewers.docx (attached as one of the submission files) and are also included below.

---

Major comment and Clinical utility:

Rigorously performed analyses demonstrating the use of advanced models. These can be used as algorithms assisting clinicians if based on few clinically available variables and highlighting certain variables of importance. The present data shows the importance of age and NPs as prognostic predictors. Can be questioned if this information adds to the clinicians’ decision and risk assessment beyond what we already know.

Response: Thank you for the comment. As the Reviewer points out, some of the identified predictors such as age and NT-proBNP have been noted previously in literature to predict outcomes in HFpEF. However, the prior existing risk models in HFpEF have been derived from registry or trial data and are for ambulatory populations. The novelty of our study is the use of real-world EHR data from a large health system and the being model specific to the hospitalized population. Thus, although there is some commonality of variables identified with prior models, the strength of association with the outcomes may differ in this treatment setting. Further, the incomplete nature of EHR data also can change association of variables. An EHR derived model is likely to perform better in a patient-facing EHR setting. However, despite these advantages further validation of our model is needed.

Specific points:

Abstract:

-

Background:

Misspelling at row 68 (HER).

Response: Thank you for pointing out this error, we have corrected it in the revision to read as ‘EHR’.

Methods:

Regarding the selection of patients based on ICD codes. Were patients with recorded LVEF =<40% in stable condition or during a later hospitalization excluded?

Was data extracted for the full time period 2008-2019?

Response: Thank you for this question. To clarify, the identification of HFpEF in this study was primarily made using ICD-9 and ICD-10 codes. However, as this is a code-based identification we wanted to enhance the validity of this method of identification by extracting ventricular function information from the associated clinical notes using regular expression.

Among the 3235 patients, 3146 (97.3%) had clinical notes available to analyze. Overall, 58.4% had an LVEF measurement value reported in the clinical notes of whom 94.0% had an LVEF value ≥ 50%, and 2.5% had an LVEF between 45 to 50%. Another 18.6% encounters had a qualitative mention of LVEF, of which 94.0% indicated the LVEF was ‘normal’ or ‘preserved’. The remaining 23% of the cohort had no quantitative or qualitative mention of ventricular information in their clinical notes.

Given that among the 77% encounters with either a quantitative or qualitative mention of ventricular mention, >97% had a documented preserved LVEF we concluded that this method of ICD code-based identification of HFpEF that we used in our study is a valid method and has sufficient positive predictive value. This validation information will also be useful for future projects by other researchers that use ICD codes to identify HFpEF.

In this process we did note 55 cases with a documented LVEF <40% and 16 cases with a qualitative mention of ‘reduced’ LVEF. However, these cases were not excluded from the cohort as the primary method of identification for this study remained ICD code-based. The notes assessment was primarily for the purpose of validation of the ICD code-based diagnosis. Hope this clarifies the authors’ query.

For vital signs and specific lab values first entry on the day of admission were used. In a clinical perspective using variables in a clinical model the last captured measurements before discharge may be more relevant.

Response: Thank you for this question. We used the vitals at admission for risk modeling as our goal was to develop a risk model that could be used at admission or in the ED, especially so for the short-term (30-day) outcome. However, for the longer-term outcome (1 year) one could potentially use either the admission values or discharge values based on when the prognostication needs to be done. For an acute hospitalization there is some utility of using admission values, such as in the well-known GRACE model for acute coronary syndrome which predicts 6 month mortality and inputs admission vitals. However, the Reviewer’s suggestion is also very valid that for longer term outcomes a model at discharge has additional utility in prognostication for patients being discharged. Our model does not fill that gap and future work could focus on a discharge-specific model for HFpEF for long-term prognostication.

Logarithmic transformations were used for certain “variables with wide ranges”. Also for variables with non-normal distribution?

Response: Thank you for the comment regarding variable transformations. We initially referred to “logarithmic transformations” in a broad sense, however, our actual preprocessing pipeline uses power and quantile transformations to handle skewed and wide-ranging variables, thereby improving the performance of models sensitive to feature scaling (Elastic Net, Lasso, Logistic Regression, SVC, and XGBoost). Random Forest and HistGradientBoosting were trained on the original features, as tree-based methods are robust to scaling. Specifically, we applied PowerTransformer (Yeo–Johnson) and QuantileTransformer (normal output) to variables such as creatinine, INR, platelet count, WBC count, and oxygen saturation, and standardized BMI to zero mean and unit variance.

Line 173 states gender was one-hot encoded – do authors refer to sex (binary)? Gender consists of >25 different categories.

Response: Thank you for pointing this error out. MIMIC records the variable sex. We have made corrections to reflect this in the text as well as Table 1.

Consider including eGFR in models instead of creatinine

Response: Thank you for this suggestion. Unfortunately with the depth of analysis that has gone into model development any change of variable entails a re-do of the entire analysis including re-calibration of the models. Other models in the HF domain such as the Get With The Guidelines (GWTG) model for HFrEF (https://pubmed.ncbi.nlm.nih.gov/24621877/) and the PREDICT-HFpEF model in HFpEF which used the DELIVER trial data (https://pubmed.ncbi.nlm.nih.gov/38536153/) also used serum creatinine as a predictor similar to us. We hope the Reviewer is OK if we keep the current model with use of Creatinine instead of eGFR given scope of work needed to change this.

Results:

Table 1

How was renal disease defined? ICD codes or per eGFR?

Response: Pre-existing renal disease in Table 1 was identified using ICD codes. Among the 36 the candidate feature as listed in Supplementary Table 3 both Prior Renal disease as identified by ICD codes as well as admission Creatinine values were used in model development.

Treatment should be of interest to include in Table 1, even if not included in models.

We were unsure what the Reviewer meant by ‘treatments’.

In terms of pre-hospital medications, this information is only available for 40.68% of the cohort through the medication reconciliation files.

The in-hospital treatments are available in detail only for those admissions that were to the ICU (17.65%).

Discharge medication list is available for the entire cohort who was discharged alive and we have added this to Table 1.

Fig 1, 2 and S2

Are in low solution and hard to read due to the small font.

Response: Thank you for pointing this out. We have generated higher resolution figures using NAAS, which accompany the revised manuscript. The submission PDF may still display the figures in lower resolution; however, the original source files contain higher-quality images for your review.

Discussion:

A discussion on how this kind of models can be used in clinical practice would add to the manuscript.

Response: Thank you for the comment. We have added the potential uses of our model in the discussion with the following lines:

‘The potential uses of these models could be early risk stratification for in-hospital planning such as ICU triage, and triggering care pathways like advanced HF team involvement, palliative care involvement etc. There is also a role for quality metrics and for hospitals to track post-discharge outcomes.’

---

## [Decision Letter · Decision Letter 1]

22 Sep 2025

Dear Dr. Murugiah,

Thank you for submitting your manuscript to PLOS ONE. After careful consideration, we feel that it has merit but does not fully meet PLOS ONE’s publication criteria as it currently stands. Therefore, we invite you to submit a revised version of the manuscript that addresses the points raised during the review process.

We look forward to receiving your revised manuscript.

Kind regards,

Shukri AlSaif

Academic Editor

PLOS ONE

Journal Requirements:

Reviewers' comments:

Reviewer's Responses to Questions

**Comments to the Author**

Reviewer #1: (No Response)

2. Is the manuscript technically sound, and do the data support the conclusions?

Reviewer #1: Yes

3. Has the statistical analysis been performed appropriately and rigorously?

Reviewer #1: Yes

4. Have the authors made all data underlying the findings in their manuscript fully available?

Reviewer #1: Yes

5. Is the manuscript presented in an intelligible fashion and written in standard English?

Reviewer #1: (No Response)

Reviewer #1: Thank you for the nicely revised manuscript. Many of the suggested changed/clarifications have been responded to, however not implemented in the ms. Clarifications below expressed as a response to reviewer may be included in the manuscript. Difficult to track in the manuscript as they are not highlighted.

Major comment and Clinical utility:

Can be questioned if this information adds to the clinicians’ decision and risk assessment beyond what we already know.

– Has to some extent been addressed in the discussion the revised ms.

Specific points:

Methods:

Regarding the selection of patients based on ICD codes. Were patients with recorded LVEF =<40% in stable condition or during a later hospitalization excluded?

-Thank you for the authors response and clarification which should be added in the ms.

-Pending: Was data extracted for the full time period 2008-2019?

Logarithmic transformations were used for certain “variables with wide ranges”. Also for variables with non-normal distribution?

-Thank you for the authors response and clarification which should be added in the ms.

Results:

Table 1

How was renal disease defined? ICD codes or per eGFR?

-Thank you for the authors response and clarification which should be added in the ms.

Treatment should be of interest to include in Table 1, even if not included in models.

-Thank you for adding treatment at discharge. Depending on data extraction period SGLT2i may be added.

Fig 1, 2 and S2

-Pending: Are in low solution and hard to read due to the small font.

**Do you want your identity to be public for this peer review?** For information about this choice, including consent withdrawal, please see our Privacy Policy

Reviewer #1: **Yes: ** Camilla Hage

---

## [Author Response · Author response to Decision Letter 2]

2 Oct 2025

Response to Reviewers:

Reviewer #1: Thank you for the nicely revised manuscript. Many of the suggested changed/clarifications have been responded to, however not implemented in the ms. Clarifications below expressed as a response to reviewer may be included in the manuscript. Difficult to track in the manuscript as they are not highlighted.

We are happy that the Reviewer found our responses satisfactory. We apologize that we did not make more detailed additions to the manuscript. In this manuscript version we have attempted to add in some more of the response letter content and highlighted the new changes appropriately.

Major comment and Clinical utility:

Can be questioned if this information adds to the clinicians’ decision and risk assessment beyond what we already know.

– Has to some extent been addressed in the discussion the revised ms.

Thank you for re-emphasizing this point. We made some additions in this version addressing this point --

“Overall, in terms of the individual risk variables most are not novel such as age and NT-proBNP and have been noted in prior studies. However, the novelty of our study remains the use of real-world EHR data from a large health system to derive the model and the being model specific to the hospitalized population. In contrast prior predictive models for HFpEF have been derived from registry or trial data and are for ambulatory populations. Thus, although there is some commonality of variables identified with prior models, the strength of association with the outcomes may differ in this treatment setting. Further, the incomplete nature of EHR data also can change the association of variables. An EHR derived model is likely to perform better in a patient-facing EHR setting. However, despite these advantages further validation of our model is needed.”

Specific points:

Methods:

Regarding the selection of patients based on ICD codes. Were patients with recorded LVEF =<40% in stable condition or during a later hospitalization excluded?

-Thank you for the authors response and clarification which should be added in the ms.

Thank you for this comment. We have added additional text addressing this point --

“Given that among the 77% encounters with either a quantitative or qualitative mention of ventricular mention, >97% had a documented preserved LVEF we concluded that this method of ICD code-based identification of HFpEF is valid and has sufficient positive predictive value. This validation exercise will also be useful for future projects by other researchers that use ICD codes to identify HFpEF. In this process we did note 55 cases with a documented LVEF <40% and 16 cases with a qualitative mention of ‘reduced’ LVEF. However, these cases were not excluded from the cohort as the primary method of identification for this study remained ICD code-based. The notes assessment was primarily for the purpose of validation of the ICD code-based diagnosis.”

-Pending: Was data extracted for the full time period 2008-2019?

Yes, correct. To clarify again these are admissions with a primary diagnosis of HFpEF. Of 430,852 total hospitalizations in MIMIC-IV data we identified 3,235 hospitalizations with a primary diagnosis of HFpEF. This prevalence is consistent with national U.S data. Around 3% of all U.S hospitalizations are for a primary diagnosis of HF. HFpEF accounts for 30-40% of all HF (ref). Secondary diagnosis codes were not used for case identification. Thus cases of HFpEF occurring with a primary diagnosis coded as pneumonia or COPD exacerbation are not counted.

We have added this time period in the study population paragraph so this is clear.

“From a total of 430,852 hospitalizations we identified 3,235 individual hospitalization encounters with a discharge diagnosis of HFpEF which comprised the study sample.”

Ref: Russo, C. A. (Thomson Medstat), Ho, K. (AHRQ), and Elixhauser, A.(AHRQ). Hospital Stays for Circulatory Diseases, 2004. HCUP Statistical Brief #26. February 2007.

Logarithmic transformations were used for certain “variables with wide ranges”. Also for variables with non-normal distribution?

-Thank you for the authors response and clarification which should be added in the ms.

Thank you for this comment. We did incorporate the specific details of the preprocessing pipeline in our last revision, however, we apologize that they were not appropriately marked as a tracked change. We have now clearly marked this in the tracked changes version. We have also made minor additions for clarification and detail --

“Based on visual inspection of the variable distributions, we applied PowerTransformer (Yeo–Johnson) or QuantileTransformer (normal output) transformations to variables with wide ranges or evident non-normal distributions (e.g., creatinine, INR, platelet count, WBC count, and oxygen saturation), while BMI was standardized to zero mean and unit variance.”

Results:

Table 1

How was renal disease defined? ICD codes or per eGFR?

-Thank you for the authors response and clarification which should be added in the ms.

Thank you for your comment. We agree that it is helpful to clarify this in the manuscript. As we mentioned previously, among the 36 candidate features in Supplementary Table 3 both Prior Renal disease as identified by ICD codes as well as admission Creatinine values were used in model development. Certainly prior renal disease affects admission creatinine, but there is additional acuity information in admission creatinine, and thus we included both as candidate variables.

To clarify this we have added the following to the ‘Methods’ section ‘Feature Selection’ paragraph --

“Note that prior Renal disease as identified by ICD codes and admission Creatinine values were both used as individual variables in model development.”

Treatment should be of interest to include in Table 1, even if not included in models.

-Thank you for adding treatment at discharge. Depending on data extraction period SGLT2i may be added.

The use of SGLT2 inhibitors was very low in the study cohort, likely as it precedes the approval of SGLT2 inhibitors for HFpEF (FDA approval in February 2022 for empagliflozin). Thus, if the Reviewer approves, we would prefer to omit this from Table 1.

Fig 1, 2 and S2

-Pending: Are in low solution and hard to read due to the small font.

We apologize for this issue. We have generated figures 1 and 2 again at higher resolution. For figure S2 we removed the variables which have mutual information scores of zero to improve readability and increased the resolution. Hopefully they meet the journal standards now.

---

## [Decision Letter · Decision Letter 2]

30 Oct 2025

Predicting 30-day and 1-year mortality in heart failure with preserved ejection fraction (HFpEF)

PONE-D-25-18607R2

Dear Dr.Karthik Murugiah,

We’re pleased to inform you that your manuscript has been judged scientifically suitable for publication and will be formally accepted for publication once it meets all outstanding technical requirements.

Kind regards,

Shukri AlSaif

Academic Editor

PLOS ONE

Reviewer #1: All comments have been addressed

2. Is the manuscript technically sound, and do the data support the conclusions?

Reviewer #1: Yes

3. Has the statistical analysis been performed appropriately and rigorously?

Reviewer #1: Yes

4. Have the authors made all data underlying the findings in their manuscript fully available?

Reviewer #1: Yes

5. Is the manuscript presented in an intelligible fashion and written in standard English?

Reviewer #1: Yes

Reviewer #1: Thank you for the re-revised manuscript. Suggested changes and clarifications have been implemented and the manuscript is now acceptable for publication

---

## [Editor Report · Acceptance letter]

PONE-D-25-18607R2

PLOS ONE

Dear Dr. Murugiah,

I'm pleased to inform you that your manuscript has been deemed suitable for publication in PLOS ONE. Congratulations! Your manuscript is now being handed over to our production team.

Kind regards,

on behalf of

Dr. Shukri AlSaif

Academic Editor

PLOS ONE